# Deep Reinforcement and InfoMax Learning

**Bogdan Mazoure**[1][*]
McGill University, Mila

**Rémi Tachet des Combes**[*]
Microsoft Research Montréal

**Thang Doan**
McGill University, Mila

**Philip Bachman**
Microsoft Research Montréal

**R Devon Hjelm**
Microsoft Research Montréal
Université de Montréal, Mila

## Abstract

We begin with the hypothesis that a model-free agent whose representations are predictive of properties of future states (beyond expected rewards) will be more capable of solving and adapting to new RL problems. To test that hypothesis, we introduce an objective based on Deep InfoMax (DIM) which trains the agent to predict the future by maximizing the mutual information between its internal representation of successive timesteps. We test our approach in several synthetic settings, where it successfully learns representations that are predictive of the future. Finally, we augment C51, a strong RL baseline, with our temporal DIM objective and demonstrate improved performance on a continual learning task and on the recently introduced Procgen environment.

## 1 Introduction

In reinforcement learning (RL), model-based agents are characterized by their ability to predict future states and rewards based on past states and actions [Sutton and Barto, 1998, Ha and Schmidhuber, 2018, Hafner et al., 2019a]. Model-based methods can be seen through the *representation learning* [Goodfellow et al., 2017] lens as endowing the agent with internal representations that are predictive of the future conditioned on its actions. This ultimately gives the agent means to plan – by e.g. considering a distribution of possible future trajectories and picking the best course of action.

In contrast, model-free methods do not explicitly model the environment, and instead learn a policy that maximizes reward or a function that estimates the optimal values of states and actions [Mnih et al., 2015, Schulman et al., 2017, Pong et al., 2018]. They can use large amounts of training data and excel in high-dimensional state and action spaces. However, this is mostly true for fixed reward functions; despite success on many benchmarks, model-free agents typically generalize poorly when the environment or reward function changes [Farebrother et al., 2018, Tachet des Combes et al., 2018] and can have high sample complexity.

Viewing model-based agents from a representation learning perspective, a desired outcome is an agent that understands the underlying generative factors of the environment that determine the observed state/action sequences, leading to generalization to other environments built from the same generative factors. In addition, learning a predictive model affords a richer learning signal than those provided by reward alone, which could reduce sample complexity compared to model-free methods.

Our work is based on the hypothesis that a model-free agent whose representations are predictive of properties of future states (beyond expected rewards) will be more capable of solving and adapting to new RL problems and, in a way, incorporate aspects of model-based learning. To learn representations

---

[*]Equal contribution. [1]Work done during an internship at Microsoft Research Montréal. Correspondence to: `bogdan.mazoure@mail.mcgill.ca`

with model-like properties, we consider a self-supervised objective derived from variants of Deep InfoMax [DIM, Hjelm et al., 2018, Bachman et al., 2019, Anand et al., 2019]. We expect this type of contrastive estimation [Hyvarinen and Morioka, 2016] will give the agent a better understanding of the underlying factors of the environment and how they relate to its actions, eventually leading to better performance in transfer and lifelong learning problems. We examine the properties of the learnt representations in simple domains such as disjoint and glued Markov chains, and more complex environments such as a 2d Ising model, a sequential variant of Ms. PacMan from the Atari Learning Environment [ALE, Bellemare et al., 2013], and all 16 games from the Procgen suite [Cobbe et al., 2019]. Our contributions are as follows:

- We propose a simple auxiliary objective that maximizes concordance between representations of successive states, given the action. We also introduce a simple adaptive mechanism that adjusts the time-scales of the contrastive tasks based on the likelihood of subsequent actions under the current RL policy.

- We present a series of experiments showing how our objective can be used as a measure of similarity and predictability, and how it behaves in partially deterministic systems.

- Finally, we show that augmenting a standard RL agent with our contrastive objective can i) lead to faster adaptation in a continual learning setting, and ii) improve overall performance on the Procgen suite.

## 2 Background

Just as humans are able to retain old skills when taught new ones [Wixted, 2004], we strive for RL agents that are able to adapt quickly and reuse knowledge when presented a sequence of different tasks with variable reward functions. The reason for this is that real-world applications or downstream tasks can be difficult to anticipate before deployment, particularly with complex environments involving other intelligent agents such as humans. Unfortunately, this proves to be very challenging even for state-of-the-art systems [Atkinson et al., 2018], leading to complex deployment scenarios.

Continual Learning (CL) is a learning framework meant to benchmark an agent's ability to adapt to new tasks by using auxiliary information about the relatedness across tasks and timescales [Kaplanis et al., 2018, Mankowitz et al., 2018, Doan et al., 2020]. Meta-learning [Thrun and Pratt, 1998, Finn et al., 2017] and multi-task learning [Hessel et al., 2019, D'Eramo et al., 2019] have shown good performance in CL by explicitly training the agent to transfer well between tasks.

In this study, we focus on the following inductive bias: while the reward function may change or vary, the underlying environment dynamics typically do not change as much[2]. To test if that inductive bias is useful, we use *auxiliary loss functions* to encourage the agent to learn about the underlying generative factors and their associated dynamics in the environment, which can result in better sample efficiency and transfer capabilities (compared to learning from rewards only). Previous work has shown this idea to be useful when training RL agents: e.g., Jaderberg et al. [2016] train the agent to predict future states given the current state-action pair, while Mohamed and Rezende [2015] uses empowerment to measure concordance between a sequence of future actions and the end state. Recent work such as DeepMDP [Gelada et al., 2019] uses a latent variable model to represent transition and reward functions in a high-dimensional abstract space. In model-based RL, various agents, such as PlaNet [Hafner et al., 2019b], Dreamer [Hafner et al., 2019a], or MuZero [Schrittwieser et al., 2019], have also shown strong asymptotic performance.

Contrastive representation learning methods are based on training an encoder to capture information that is shared across different views of the data in the features it produces for each input. The similar (i.e. positive) examples are typically either taken from different "locations" of the data [e.g., spatial patches or temporal locations, see Hjelm et al., 2018, Oord et al., 2018, Anand et al., 2019, Hénaff et al., 2019] or obtained through data augmentation [Wu et al., 2018, He et al., 2019, Bachman et al., 2019, Tian et al., 2019, Chen et al., 2020]. Contrastive models rely on a variety of objectives to encourage similarity between features. Typically, a scoring function [e.g., dot product or cosine similarity between pairs of features, see Wu et al., 2018] that lower-bounds mutual information is maximized [Belghazi et al., 2018, Hjelm et al., 2018, Oord et al., 2018, Poole et al., 2019].

A number of works have applied the above ideas to RL settings. Contrastive Predictive Coding [CPC, Oord et al., 2018] augments an A2C agent with an autoregressive contrastive task across a sequence of frames, improving performance on 5 DeepMind lab games [Beattie et al., 2016]. EMI [Kim et al., 2019] uses a Jensen-Shannon divergence-based lower bound on mutual information across subsequent frames as an exploration bonus. CURL [Srinivas et al., 2020] uses a contrastive task using augmented versions of the same frame (does not use future frames) as an auxiliary task to an RL algorithm. Finally, HOMER [Misra et al., 2019] produces a policy cover for block MDPs by learning backward and forward state abstractions using contrastive learning objectives. It is worth noting that HOMER has statistical guarantees for its performance on certain hard exploration problems.

Our work, DRIML, predicts future states conditioned on the current state-action pair at multiple scales, drawing upon ideas encapsulated in Augmented Multiscale Deep InfoMax [AMDIM, Bachman et al., 2019] and Spatio-Temporal DIM [ST-DIM, Anand et al., 2019]. Our method is flexible w.r.t. these tasks: we can employ the DIM tasks over features that constitute the full frame (global) or that are specific to local patches (local) or both. It is also robust w.r.t. time-scales of the contrastive tasks, though we show that adapting this time scale according to the predictability of subsequent actions under the current RL policy improves performance substantially.

## 3 Preliminaries

We assume the usual Markov Decision Process (MDP) setting (see Appendix for details), with the MDP denoted as $\mathcal{M}$, states as $s$, actions as $a$, and the policy as $\pi$. Since we focus on exploring the role of auxiliary losses in continuous learning, we use C51 [Bellemare et al., 2017], which extends DQN [Mnih et al., 2015] to predict the full distribution of potential future rewards, for training the agent due to its strong performance on control tasks from pixels. C51 minimizes the following loss:

$$\mathcal{L}_{RL} = D_{KL}(\lfloor \mathcal{T}Z(s,a) \rfloor_{51} || Z(s,a)), \tag{1}$$

where $D_{KL}$ is the Kullback-Leibler divergence, $Z(s,a)$ is the distribution of future discounted returns under the current policy ($\mathbb{E}[Z(s,a)] = Q(s,a)$), $\mathcal{T}$ is the distributional Bellman operator [Bellemare et al., 2019] and $\lfloor \cdot \rfloor_{51}$ is an operator which projects $Z$ onto a fixed support of 51.

### 3.1 State-action mutual information maximization

Mutual information (MI) measures the amount of information shared between a pair of random variables and can be estimated using neural networks [Belghazi et al., 2018]. Recent representation learning algorithms [Oord et al., 2018, Hjelm et al., 2018, Tian et al., 2019, He et al., 2019] train encoders to maximize the MI between features taken from different views of the input – e.g., different patches in an image, different timesteps in a sequence, or different versions of an image produced by applying data augmentation to it.

Let $k$ be some fixed temporal offset. Running a policy $\pi$ in the MDP $\mathcal{M}$ generates a distribution over tuples $(s_t, a_t, s_{t+k})$, where $s_t$ corresponds to the state of $\mathcal{M}$ at some timestep $t$, $a_t$ to the action selected by $\pi$ in state $s_t$ and $s_{t+k}$ to the state of $\mathcal{M}$ at timestep $t+k$, reached by following $\pi$. $S_t$, $A_t$ and $S_{t+k}$ stand for the corresponding random variables. We also denote the joint distribution of these variables, as well as their associated marginals, using $p$. We are interested in learning representations of state-action pairs that have high MI with the representation of states later in the trajectory. The MI between e.g. state-action pairs $(S_t, A_t)$ and their future states $S_{t+k}$ is defined as follows:

$$\mathcal{I}([S_t, A_t], S_{t+k}; \pi) = \mathbb{E}_{p_\pi(s_t, a_t, s_{t+k})} \left[ \log \frac{p_\pi(s_t, a_t, s_{t+k})}{p_\pi(s_t, a_t) p_\pi(s_{t+k})} \right], \tag{2}$$

where $p_\pi$ denotes distributions under $\pi$. Estimating the MI can be done by training a classifier that discriminates between a sample drawn from the joint distribution – the numerator of Eq. 2 – and a sample from the product of marginals – its denominator. A sample from the product of marginals is usually obtained by replacing $s_{t+k}$ (positive sample) with a state picked at random from another trajectory (negative sample). Letting $S^-$ denote a set of such negative samples, the infoNCE loss function [Gutmann and Hyvärinen, 2010, Oord et al., 2018] that we use to maximize a lower bound on the MI in Eq. 2 (with the added encoders for the states and actions) takes the following form:

$$\mathcal{L}_{NCE} := -\mathbb{E}_{p_\pi(s_t, a_t, s_{t+k})} \mathbb{E}_{S^-} \left[ \log \frac{\exp(\phi(\Psi(s_t, a_t), \Phi(s_{t+k})))}{\sum_{s' \in S^- \cup \{s_{t+k}\}} \exp(\phi(\Psi(s_t, a_t), \Phi(s')))} \right], \tag{3}$$

where $\Psi(s, a), \Phi(s)$ are features that depend on state-action pairs and states, respectively, and $\phi$ is a function that outputs a scalar-valued *score*. Minimizing $\mathcal{L}_{NCE}$ with respect to $\Phi, \Psi$, and $\phi$ maximizes the MI between these features. In practice, we construct $S^-$ by including all states $\tilde{s}_{t+k}$ from other tuples $(\tilde{s}_t, \tilde{a}_t, \tilde{s}_{t+k})$ in the same minibatch as the relevant $(s_t, a_t, s_{t+k})$. I.e., for a batch containing $N$ tuples $(s_t, a_t, s_{t+k})$, each $S^-$ would contain $N-1$ negative samples.

## 4 Architecture and Algorithm

We now specify forms for the functions $\Phi, \Psi$, and $\phi$. We consider a deep neural network $\Theta : \mathcal{S} \to \prod_{i=1}^{5} \mathcal{F}_i$ which maps input states onto a sequence of progressively more "global" (or less "local") feature spaces. In practice, $\Theta$ is a CNN composed of functions that sequentially map inputs to features $\{f_i \in \mathcal{F}_i\}_{1 \leq i \leq 5}$ (lower to upper "levels" of the network). For ease of explanation, we formulate our model using specific features (e.g., *local* features $f_3$ and *global* features $f_4$), but our model covers any set of features extracted from $\Theta$ used for the objective below as well as other choices for $\Theta$.

The features $f_5$ are the output of the network's last layer and correspond to the standard C51 value heads (i.e., they span a space of 51 atoms per action) [3]. For the auxiliary objective, we follow a variant of Deep InfoMax [DIM, Hjelm et al., 2018, Anand et al., 2019, Bachman et al., 2019], and train the encoder to maximize the mutual information (MI) between local and global "views" of tuples $(s_t, a_t, s_{t+k})$. The local and global views are realized by selecting $f_3 \in \mathcal{F}_3$ and $f_4 \in \mathcal{F}_4$ respectively. In order to simultaneously estimate and maximize the MI, we embed the action (represented as a one-hot vector) using a function $\Psi_a : \mathcal{A} \to \tilde{\mathcal{A}}$. We then map the local states $f_3$ and the embedded action using a function $\Psi_3 : \mathcal{F}_3 \times \tilde{\mathcal{A}} \to L$, and do the same with the global states $f_4$, i.e., $\Psi_4 : \mathcal{F}_4 \times \tilde{\mathcal{A}} \to G$. In addition, we have two more functions, $\Phi_3 : \mathcal{F}_3 \to L$ and $\Phi_4 : \mathcal{F}_4 \to G$ that map features without the actions, which will be applied to features from "future" timesteps. Note that $L$ can be thought of as a product of local spaces (corresponding to different patches in the input, or equivalently different receptive fields), each with the same dimensionality as $G$.

We use the outputs of these functions to produce a scalar-valued score between any combination of local and global representations of state $s_t$ and $s_{t+k}$, conditioned on action $a_t$:

$$\phi_{NtM}(s_t, a, s_{t+k}) := \Psi_N(f_N(s_t), \Psi_a(a_t))^\top \Phi_M(f_M(s_{t+k})), \ M, N \in \{3, 4\}. \qquad (4)$$

In practice, for the functions that take features and actions as input, we simply concatenate the values at position $f_3$ (local) or $f_4$ (global) with the embedded action $\Psi_a(a)$, and feed the resulting tensor into the appropriate function $\Psi_3$ or $\Psi_4$. All functions that process global and local features are computed using $1 \times 1$ convolutions. See Figure 4 for a visual representation of our model.

We use the scores from Eq. 4 when computing the infoNCE loss [Oord et al., 2018] for our objective, using $(s_t, a_t, s_{t+k})$ tuples sampled from trajectories stored in an experience replay buffer:

$$\mathcal{L}_{DIM}^{NtM} := -\mathbb{E}_{p_\pi(s_t, a_t, s_{t+k})} \mathbb{E}_{S^-} \left[ \log \frac{\exp(\phi_{NtM}(s_t, a_t, s_{t+k}))}{\sum_{s' \in S^- \cup \{s_{t+k}\}} \exp(\phi_{NtM}(s_t, a_t, s'))} \right]. \qquad (5)$$

Combining Eq. 5 with the RL update in Eq. 1 yields our full training objective, which we call DRIML [4]. We optimize $\Theta, \Psi_{3,4,a}$, and $\Phi_{3,4}$ jointly using a single loss function:

$$\mathcal{L}_{DRIML} = \mathcal{L}_{RL} + \sum_{M,N \in \{3,4\}} \lambda_{NtM} \mathcal{L}_{DIM}^{NtM} \qquad (6)$$

Note that, in practice, the compute cost which Eq. 6 adds to the core RL algorithm is minimal, since it only requires additional passes through the (small) state/action embedding functions followed by an outer product.

---

**Algorithm 1:** Deep Reinforcement and InfoMax Learning (DRIML)

---

**Input** : Batch $\mathcal{B}$ sampled from the replay buffer, $\{\lambda_{NtM}\}_{N,M\in\{3,4\}}$, strictly positive integer $k$
Update $\Theta$ using Eq. 1;
$s, a, s', x \leftarrow \mathcal{B}[s_t], \mathcal{B}[a_t], \mathcal{B}[s_{t+k}], \mathcal{B}[s_{t'\neq t+k}]$;
**for** $N$ in $\{3,4\}$ **do**
    **for** $M$ in $\{3,4\}$ **do**
        **if** $\lambda_{NtM} > 0$ **then**
            Compute $\mathcal{L}_{DIM}^{NtM}$ using Eq. 5 (see Appendix 8.5 for PyTorch code);
            Update $\Theta$, $\Psi_{3,4,a}$, and $\Phi_{3,4}$ using gradients of $\lambda_{NtM}\mathcal{L}_{DIM}^{NtM}$;
        **end**
    **end**
**end**

---

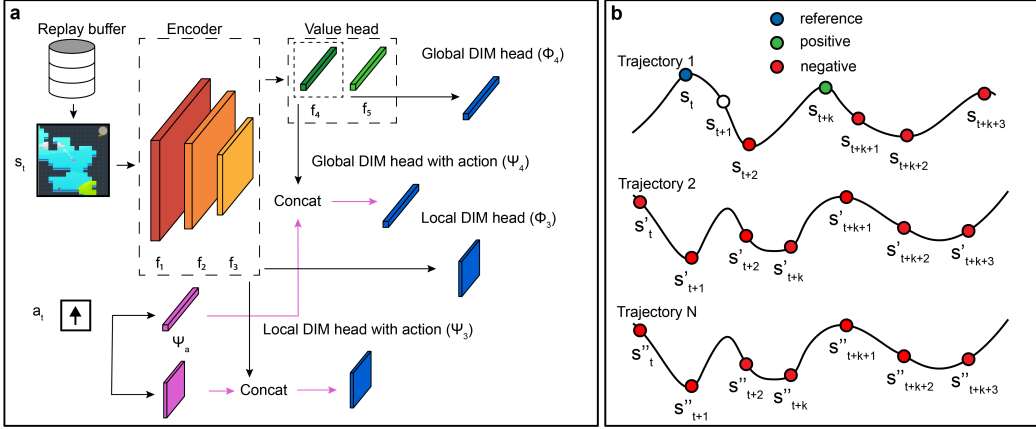

Figure 1: **(a)** Example model architecture used for the encoder used for the RL and DIM objectives and **(b)** distribution of reference, positive and negative samples within training batch $\mathcal{B}$. Note that in our experiments, we either use only the local head, only the global head, or both.

The proposed Algorithm 1 introduces an auxiliary loss which improves predictive capabilities of value-based agents by boosting similarity of representations close in time.

## 5 Finding the Best Task Timescale

The above DRIML algorithm fixes the temporal offset for the contrastive task, $k$, which needs to be chosen a-priori. However, different games are based on MDPs whose dynamics operate at different timescales, which in turn means that the difficulty of predictive tasks across different games will vary at different scales. We could predict simultaneously at multiple timescales [as in Oord et al., 2018], yet this introduces additional complexity that could be overcome by simply finding the *right* timescale. In order to ensure our auxiliary loss learns useful insights about the underlying MDP, as well as make DRIML more generally useful across environments, we adapt the temporal offset $k$ automatically based on the distribution of the agent's actions.

We propose to select an adaptive, game-specific look-ahead value $k$, by learning a function $q_\pi(a_i, a_j)$ which measures the log-odds of taking action $a_j$ after taking action $a_i$ when following policy $\pi$ in the environment (i.e. $p_\pi(A_{t+1} = a_j | A_t = a_i)/p_\pi(A_{t+1} = a_j)$). The values $q_\pi(a_i, a_j)$ are then used to sample a look-ahead value $k \in \{1, .., H\}$ from a non-homogeneous geometric (NHG) distribution. This particular choice of distribution was motivated by two desirable properties of NHG: **(a)** any

discrete positive random variable can be represented via NHG [Mandelbaum et al., 2007] and **(b)** the expectation of $X \sim NHG(q_1, .., q_H)$ obeys the rule $1/\max_i q_i \leq \mathbb{E}[X] \leq 1/\min_i q_i$. The intuition is that, if the state dynamics are regular, this should be reflected in the predictability of the future actions conditioned on prior actions. Our algorithm captures this through a matrix $\mathbf{A}$, whose $i$-th row is the softmax of $q_\pi(a_i, \cdot)$. $q_\pi(a_i, a_j)$ is learned off-policy using the same data from the buffer as the main algorithm; it is updated at the same frequency as the main DRIML parameters and trained to approximate the relevant log odds. This modification is small in relation to the DRIML algorithm, but it substantially improves results in our experiments. The sampling of $k$ is done via Algorithm 2, and additional analysis of the adaptive temporal offset is provided in Figure 2.

---

**Algorithm 2:** Adaptive lookahead selection

---

**Input**    : Tuple $(s_t, a_t, a_{t+1}, ..., a_{t+H})$, maximal horizon $H$, stochastic matrix $\mathbf{A}$ of size $\mathcal{A} \times \mathcal{A}$
**Output** : Lookahead value $k : 1 \leq k \leq H$
**for** $i$ *in* $1, ..., H$ **do**
    $k \leftarrow i$; // Updating value of k
    $b \sim \text{Bernoulli}(\mathbf{A}_{a_{t+i-1}, a_{t+i}})$;
    **if** $b == 0$ **then**
        | break;
    **end**
**end**

---

Figure 2 shows the impact of adaptively selecting $k$ using the NHG sampling method. For instance, (i) depending on the nature of the game, DRIML-ada tends to repeat movement actions in navigation games and repeatedly fire in shooting games, and (ii) the value of $k$ tends to converge to 1 for games like Bigfish and Plunder as training progresses, which hints to an exploration-exploitation like trade-off.

Since many Procgen games do not have special actions such as fire or diagonal moves, DRIML-ada considers the actual actions (15 of them) and the visible actions (at most 15 of them) together in the adaptive lookahead selection algorithm.

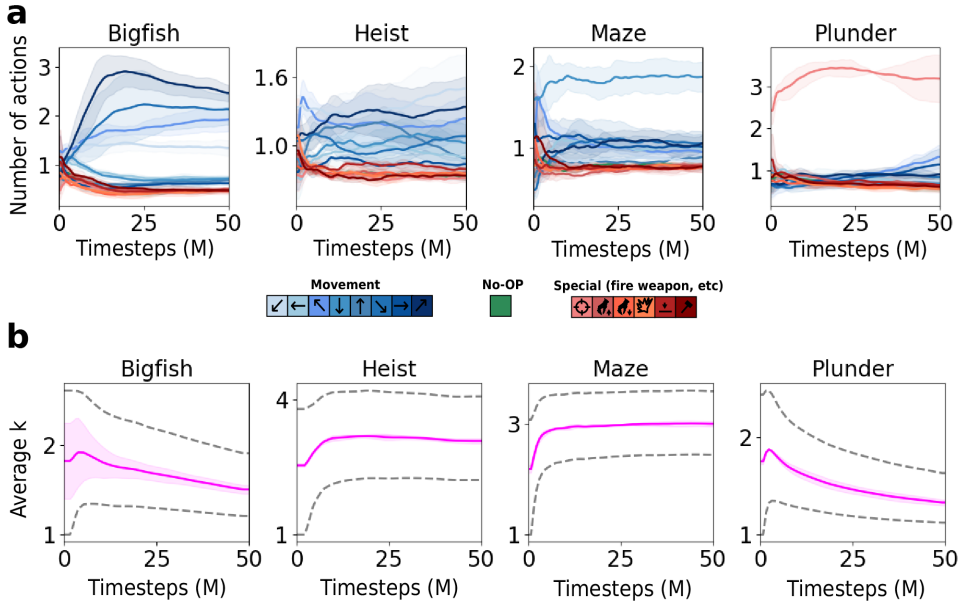

Figure 2: **(a)** Average number of movement, no-op and special actions taken by DRIML-ada on 4 Procgen games and **(b)** change in the average, max and min across batch values of $k$ as a function of training steps.

# 6 Experiments

In this section, we first show how our proposed objective can be used to estimate state similarity in single Markov chains. We then show that DRIML can capture dynamics in locally deterministic systems (Ising model), which is useful in domains with partially deterministic transitions. We then provide results on a continual version of the Ms. PacMan game where the DIM loss is shown to converge faster for more deterministic tasks, and to help in a continual learning setting. Finally, we provide results on Procgen [Cobbe et al., 2019], which show that DRIML performs well when trained on 500 levels with fixed order. All experimental details can be found in Appendix 8.6.

## 6.1 DRIML learns a transition ratio model

We first study the behaviour of DRIML's loss on a simple Markov chain describing a biased random walk in $\{1, \cdots, K\}$. The bias is specified by a single parameter $\alpha$. The agent starting at state $i$ transitions to $i+1$ with probability $\alpha$ and to $i-1$ otherwise. The agent stays in states $1$ and $K$ with probability $1-\alpha$ and $\alpha$, respectively. We encode the current and next states (represented as one-hots) using a 1-hidden layer MLP[5] (corresponding to $\Psi$ and $\Phi$ in equation 3), and then optimize the NCE loss $\mathcal{L}_{DIM}$ (the scoring function $\phi$ is also 1-hidden layer MLP, equation 3) to maximize the MI between representations of successive states. Results are shown in Fig. 3b, they are well aligned with the true transition matrix (Fig. 3c).

This toy experiment revealed an interesting observation: DRIML's objective involves computing the ratio of the Markov transition operator over the stationary distribution, implying that the convergence rate is affected by the *spectral gap* of the average Markov transition operator, $\mathbf{T}^\pi_{ss'} = \mathbb{E}_{a \sim \pi(s, \cdot)}[T(s, a, s')]$ for transition operator $T$. That is, it depends on the difference between the two largest eigenvalues of $\mathbf{T}^\pi$, namely $1$ and $\lambda_{(2)}$. In the case of the random walk, the spectral gap of its transition matrix can be computed in closed-form as a function of $\alpha$. Its lowest value is reached in the neighbourhood $\alpha = 0.5$, corresponding to the point where the system is least predictable (as shown by the mutual information, Fig 3c). However, since the matrix is not invertible for $\alpha = 0.5$, we consider $\alpha = 0.499$ instead. Derivations are available in Appendix 8.6.1, and more insights on the connection to the spectral gap are presented in Appendix 8.4.

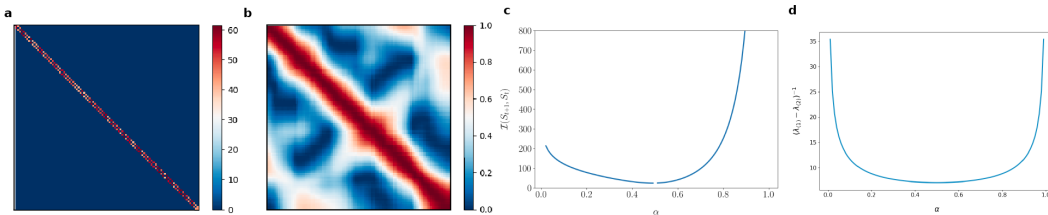

Figure 3: **(a)** Ratio of transition matrix over stationary vector for the random walk with $\alpha = 0.499$, **(b)** the prediction matrix of being a pair of successive states learnt by $\mathcal{L}_{DIM}$, **(c)** the closed-form mutual information between consecutive states in time as a function of $\alpha$ (with simplified endpoint conditions) and **(d)** the true inverse spectral gap $(\lambda_{(1)} - \lambda_{(2)})^{-1}$ as a function of $\alpha$.

## 6.2 DRIML can capture complex partially deterministic dynamics

The goal of this experiment is to highlight the predictive capabilities of our DIM objective in a partially deterministic system. We consider a dynamical system composed of $N \times N$ pixels with values in $\{-1, 1\}$, $S(t) = \{s_{ij}(t) \mid 1 \le i, j \le N\}$. At the beginning of each episode, a patch corresponding to a quarter of the pixels is chosen at random in the grid. Pixels that do not belong to that patch evolve fully independently ($p(s_{ij}(t) = 1 \mid S(t-1)) = p(s_{ij}(t) = 1) = 0.5$). Pixels from the patch obey a local dependence law, in the form of a standard Ising model[6]: the value of a pixel at time $t$ only depends on the value of its neighbors at time $t-1$. This local dependence is obtained through a function $f$: $p(s_{ij}(t) | S(t-1)) = f(\{s_{i'j'}(t-1) \mid |i-i'| = |j-j'| = 1\})$ (see

Appx 8.6.2 for details). Figure 4 shows the system at $t = 32$ during three different episodes (black pixels correspond to values of $-1$, white to 1). The patches are very distinct from the noise. We then train a convolutional encoder using our DIM objective on local "views" only (see Section 4).

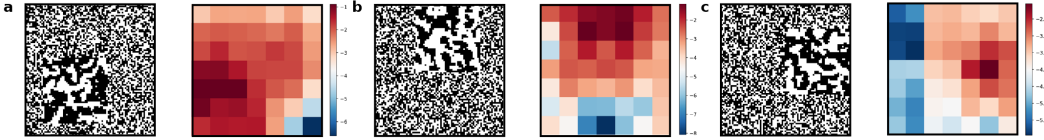

Figure 4: $42 \times 42$ Ising model with temperature $\beta^{-1} = 0.4$ overlaid onto a $84 \times 84$ lattice of uniformly random spins $\{-1, +1\}$. The grayscale plots show each of the 3 systems at $t = 32$; the color plots show the DIM similarity scores between $t = 2$ and $t = 3$.

Figure 4 shows the similarity scores between the local features of states at $t = 2$ and the same features at $t = 3$ (a local feature corresponds to a specific location in the convolutional maps)[7]. The heatmap regions containing the Ising model (larger-scale patterns) have higher scores than the noisy portions of the lattice. Local DIM is able to correctly encode regions of high temporal predictability.

## 6.3  A continual learning experiment on Ms. PacMan

We further complicate the task of the Ising model prediction by building on top of the Ms. PacMan game and introducing non-trivial dynamics. The environment is shown in the appendix.

In order to assess how well our auxiliary objective captures predictability in this MDP, we define its dynamics such that $\mathbb{P}[\text{Ghost}_i \text{ takes a random move}] = \varepsilon$. Intuitively, as $\varepsilon \to 1$, the enemies' actions become less predictable, which in turn hinders the convergence rate of the contrastive loss. The four runs in Figure 5a correspond to various values of $\varepsilon$. We trained the agent using our $\mathcal{L}_{DRIML}$ objective. We can see that the convergence of the auxiliary loss becomes slower with growing $\varepsilon$, as the model struggles to predict $s_{t+1}$ given $(s_t, a_t)$. After 100k frames, the NCE objective allows to separate the four MDPs according to their principal source of randomness (red and blue curves). When $\varepsilon$ is close to 1, the auxiliary loss has a harder time finding features that predict the next state, and eventually ignores the random movements of enemies.

The second and more interesting setup we consider consists in making only one out of 4 enemies lethal, and changing which one every 5k episodes. Figure 5b shows that, as training progresses, the blue curve (C51) always reaches the same performance at the end of the 5k episodes, while DRIML's steadily increases. The blue agent learns to ignore the harmless ghosts (they have no connection to the reward signal) and has to learn the task from scratch every time the lethal ghost changes. On the other hand, the DRIML agent (red curve) is incentivized to encode information about all the predictable objects on the screen (including the harmless ghosts), and as such adapts faster and faster to changes. Figure 5c shows the same PacMan environment with a quasi-deterministic Ising model evolving in the walled areas of the screen (details in appendix). For computational efficiency, we only run this experiment for 10k episodes. As before, DRIML outperforms C51 after the lethal ghost change, demonstrating that its representations encode more information about the dynamics of the environment (in particular about the harmless ghosts). The presence of additional distractors - the Ising model in the walls - did not impact that observation.

## 6.4  Performance on Procgen Benchmark

Finally, we demonstrate the beneficial impact of adding a DIM-like objective to C51 (DRIML) on the 500 first levels of all 16 Procgen tasks [Cobbe et al., 2019]. All algorithms are trained for 50M environment frames with the DQN [Mnih et al., 2015] architecture. The mean and standard deviation of the scores (over 3 seeds) are shown in Table 1; bold values indicate best performance.

Similarly to CURL, we used data augmentation on inputs to DRIML-fix to improve the model's predictive capabilities in fast-paced environments (see App. 8.6.4). While we used the global-global

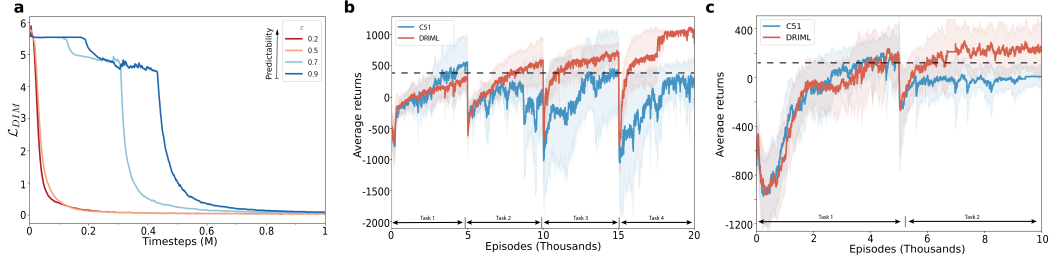

Figure 5: **(a)** average training NCE loss for various values of $\varepsilon$ as a function of timesteps, **(b)** average training reward with only one harmful enemy per level (dashed line indicates average terminal blue curve performance after each task) and **(c)** average training reward on PacMan + Ising noise in walled areas.

Table 1: Average training returns collected after 50M of training frames, $\pm$ one standard deviation.

| Env | C51 | CPC-1$\to$5 | CURL | DRIML-noact | DRIML-randk | DRIML-fix | DRIML-ada |
|---|---|---|---|---|---|---|---|
| bigfish | $1.33 \pm 0.12$ | $1.17 \pm 0.16$ | $2.70 \pm 1.30$ | $1.19 \pm 0.04$ | $1.12 \pm 1.03$ | $2.02 \pm 0.18$ | $\mathbf{4.45 \pm 0.71}$ |
| bossfight | $0.57\pm0.05$ | $0.52\pm0.07$ | $0.60\pm0.06$ | $0.47\pm0.01$ | $0.56\pm0.03$ | $0.67\pm0.02$ | $\mathbf{1.05\pm0.19}$ |
| caveflyer | $9.19\pm0.29$ | $6.40\pm0.56$ | $6.94\pm0.25$ | $8.26\pm0.26$ | $7.92\pm0.15$ | $\mathbf{10.2\pm0.41}$ | $6.77\pm0.04$ |
| chaser | $0.22\pm0.04$ | $0.21\pm0.02$ | $0.35\pm0.04$ | $0.23\pm0.02$ | $0.26\pm0.01$ | $0.29\pm0.02$ | $\mathbf{0.38\pm0.04}$ |
| climber | $1.68\pm0.10$ | $1.71\pm0.11$ | $1.75\pm0.09$ | $1.57\pm0.01$ | $2.21\pm0.48$ | $\mathbf{2.26\pm0.05}$ | $2.20\pm0.08$ |
| coinrun | $\mathbf{29.7\pm5.44}$ | $11.4\pm1.55$ | $21.2\pm1.94$ | $13.2\pm1.21$ | $21.6\pm1.97$ | $27.2\pm1.92$ | $22.88\pm0.4$ |
| dodgeball | $1.20\pm0.08$ | $1.05\pm0.04$ | $1.09\pm0.04$ | $1.22\pm0.04$ | $1.19\pm0.03$ | $1.28\pm0.02$ | $\mathbf{1.44\pm0.06}$ |
| fruitbot | $3.86\pm0.96$ | $4.56\pm0.93$ | $4.89\pm0.71$ | $5.42\pm1.33$ | $6.84\pm0.24$ | $5.40\pm1.02$ | $\mathbf{9.53\pm0.29}$ |
| heist | $1.54\pm0.10$ | $0.93\pm0.08$ | $1.06\pm0.05$ | $1.04\pm0.02$ | $1.00\pm0.05$ | $1.30\pm0.05$ | $\mathbf{1.89\pm0.02}$ |
| jumper | $\mathbf{13.2\pm0.83}$ | $2.28\pm0.44$ | $10.3\pm0.61$ | $4.31\pm0.64$ | $5.62\pm0.27$ | $12.6\pm0.64$ | $12.2\pm0.42$ |
| leaper | $5.03\pm0.14$ | $4.01\pm0.71$ | $3.94\pm0.46$ | $5.40\pm0.09$ | $4.24\pm1.17$ | $6.17\pm0.29$ | $\mathbf{6.35\pm0.46}$ |
| maze | $2.36\pm0.09$ | $1.14\pm0.08$ | $0.82\pm0.20$ | $1.44\pm0.26$ | $1.18\pm0.03$ | $1.38\pm0.08$ | $\mathbf{2.62\pm0.10}$ |
| miner | $0.13\pm0.01$ | $0.13\pm0.02$ | $0.10\pm0.01$ | $0.12\pm0.01$ | $0.15\pm0.01$ | $0.14\pm0.01$ | $\mathbf{0.19\pm0.02}$ |
| ninja | $\mathbf{9.36\pm0.01}$ | $6.23\pm0.82$ | $5.84\pm1.21$ | $6.44\pm0.22$ | $8.13\pm0.26$ | $9.21\pm0.25$ | $8.74\pm0.28$ |
| plunder | $2.99\pm0.07$ | $3.00\pm0.06$ | $2.77\pm0.14$ | $3.20\pm0.05$ | $3.34\pm0.09$ | $3.37\pm0.17$ | $\mathbf{3.58\pm0.04}$ |
| starpilot | $2.44\pm0.12$ | $2.87\pm0.05$ | $2.68\pm0.09$ | $3.70\pm0.30$ | $3.93\pm0.04$ | $\mathbf{4.56\pm0.21}$ | $2.63\pm0.16$ |
| Norm.score | 1.0 | 0.23 | 0.52 | 0.59 | 0.92 | 1.48 | 1.9 |

loss in DRIML's objective for all Procgen games, we have found that the local-local loss also had a beneficial effect on performance on a smaller set of games (e.g. starpilot, which has few moving entities on a dark background).

## 7 Discussion

In this paper, we introduced an auxiliary objective called Deep Reinforcement and InfoMax Learning (DRIML), which is based on maximizing concordance of state-action pairs with future states (at the representation level). We presented results showing that 1) DRIML implicitly learns a transition model by boosting state similarity, 2) it can improve performance of deep RL agents in a continual learning setting and 3) it boosts training performance in complex domains such as Procgen.

## Acknowledgements

We thank Harm van Seijen, Ankesh Anand, Mehdi Fatemi, Romain Laroche and Jayakumar Subramanian for useful feedback and helpful discussions.

## Broader Impact

This work proposes an auxiliary objective for model-free reinforcement learning agents. The objective shows improvements in a continual learning setting, as well as on average training rewards for a suite of complex video games. While the objective is developed in a visual setting, maximizing

mutual information between features is a method that can be transported to other domains, such as text. Potential applications of deep reinforcement learning are (among others) healthcare, dialog systems, crop management, robotics, etc. Developing methods that are more robust to changes in the environment, and/or perform better in a continual learning setting can lead to improvements in those various applications. At the same time, our method fundamentally relies on deep learning tools and architectures, which are hard to interpret and prone to failures yet to be perfectly understood. Additionally, deep reinforcement learning also lacks formal performance guarantees, and so do deep reinforcement learning agents. Overall, it is essential to design failsafes when deploying such agents (including ours) in the real world.

## Funding

BM started the work while intern at Microsoft Research, Montreal, and later received support as a graduate student fellow from FRQNT.

## Footnotes

[2]This is not true in all generalization settings. Generalization still has a variety of specifications within RL. In our work, we focus on the setting where the rewards change more rapidly than the environment dynamics.

[3] $\Theta$ can equivalently be seen as the network used in a standard C51.

[4] Deep Reinforcement and InfoMax Learning

[5]The action is simply ignored in this setting.

[6]https://en.wikipedia.org/wiki/Ising_model

[7]We chose early timesteps to make sure that the model does not simply detect large patches, but truly measures predictability.

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
