[Supplementary Material]

# 8 Appendix

## 8.1 Markov Chains

Given a discrete state space $\mathcal{S}$ with probability measure $\mathbb{T}$, a discrete-time homogeneous Markov chain (MC) is a collection of random variables with the following property on its *transition matrix* $\mathbf{T} \in \mathbb{R}^{|\mathcal{S}| \times |\mathcal{S}|}$: $\mathbf{T}_{ss'} = \mathbb{P}[S_{t+1} = s'|S_t = s]$, $\forall s, s' \in \mathcal{S}, \forall t \geq 0$. Assuming the Markov chain is ergodic, its *invariant distribution*[8] $\boldsymbol{\rho}$ is the principal eigenvector of $\mathbf{T}$, which verifies $\boldsymbol{\rho}\mathbf{T} = \boldsymbol{\rho}$ and summarizes the long-term behaviour of the chain. We define the marginal distribution of $S_t$ as $p_t(s) := \mathbb{P}[S_t = s] = \mathbf{T}_{s:}^t$, and the initial distribution of $S$ as $p_0(s)$.

## 8.2 Markov Decision Processes

A discrete-time, finite-horizon Markov Decision Process [Bellman, 1957, Puterman, 2014, MDP] comprises a state space $\mathcal{S}$, an action space[9] $\mathcal{A}$, a transition kernel $T : \mathcal{S} \times \mathcal{A} \times \mathcal{S} \mapsto [0, 1]$, a reward function $r : \mathcal{S} \times \mathcal{A} \mapsto \mathbb{R}$ and a discount factor $\gamma \in [0, 1]$. At every timestep $t$, an agent interacting with this MDP observes the current state $s_t \in \mathcal{S}$, selects an action $a_t \in \mathcal{A}$, and observes a reward $r(s_t, a_t) \in \mathbb{R}$ upon transitioning to a new state $s_{t+1} \sim T(s_t, a_t, \cdot)$. The goal of an agent in a discounted MDP is to learn a policy $\pi : \mathcal{S} \times \mathcal{A} \mapsto [0, 1]$ such that taking actions $a_t \sim \pi(\cdot|s_t)$ maximizes the expected sum of discounted returns,

$$V^{\pi}(s) = \mathbb{E}_{\pi}\left[\sum_{t=0}^{\infty} \gamma^t r(s_t, a_t)|s_0 = s\right].$$

To convert a MDP into a MC, one can let $\mathbf{T}_{ss'}^{\pi} = \mathbb{E}_{a \sim \pi(s, \cdot)}[T(s, a, s')]$, an operation which can be easily tensorized for computational efficiency in small state spaces [see Mazoure et al., 2020].

## 8.3 Link to invariant distribution

For a discrete state ergodic Markov chain specified by $\mathbf{P}$ and initial occupancy vector $\boldsymbol{p}_0$, its marginal state distribution at time $t$ is given by the Chapman-Kolmogorov form:

$$\mathbb{P}[S_t = s] = \boldsymbol{p}_0 \mathbf{P}_{\cdot s}^t, \tag{7}$$

and its *limiting distribution* $\boldsymbol{\sigma}$ is the infinite-time marginal

$$\lim_{t \to \infty} \mathbf{P}_{ss'}^{(t)} = \boldsymbol{\sigma}_{s'}, \ s, s' \in \mathcal{S} \tag{8}$$

which, if it exists, is exactly equal to the *invariant distribution* $\boldsymbol{\rho}$.

For the very restricted family of ergodic MDPs under fixed policy $\pi$, we can assume that $p_t$ converges to a time invariant distribution $\rho$.

Therefore,

$$\mathcal{I}_t(S, S') = \sum_{s' \in \mathcal{S}} \sum_{s \in \mathcal{S}} p_0(\mathbf{P}^t)_{:s} \mathbf{P}_{ss'}\left(\log \mathbf{P}_{ss'} - \log\{p_0(\mathbf{P}^{t+1})_{:s'}\}\right) \tag{9}$$

Now, observe that $\mathcal{I}_t$ is closely linked to $T/\rho$ when samples come from timesteps close to $t_{mix}(\varepsilon)$. That is, interchanging swapping $\rho(s)$ and $p_t(s)$ at any state $s$ would yield at most $\delta(t)$ error. Moreover, existing results [Levin and Peres, 2017] from Markov chain theory provide bounds on $||(\mathbf{P}^{t+1})_{s:} - (\mathbf{P}^t)_{s:}||_{TV}$ depending on the structure of the transition matrix.

If $\mathbf{P}$ has a limiting distribution $\boldsymbol{\sigma}$, then using the dominated convergence theorem allows to replace matrix powers by $\boldsymbol{\sigma}$, which is then replaced by the invariant distribution $\boldsymbol{\rho}$:

$$\lim_{t \to \infty} \mathcal{I}_t = \sum_{s' \in \mathcal{S}} \sum_{s \in \mathcal{S}} \boldsymbol{\rho}_s \boldsymbol{\rho}_{s'}\left(\log \mathbf{P}_{ss'} - \log \boldsymbol{\rho}_{s'}\right)$$
$$= \mathbb{E}_{\rho \times \rho}\left(\log \mathbf{P}_{ss'} - \log \boldsymbol{\rho}_{s'}\right) \tag{10}$$

Of course, most real-life Markov decision processes do not actually have an invariant distribution since they have absorbing (or terminal) states. In this case, as the agent interacts with the environment, the DIM estimate of MI yields a rate of convergence which can be estimated based on the spectrum of $\mathbf{P}$.

Moreover, one could argue that since, in practice, we use off-policy algorithms for this sort of task, the gradient signal comes from various timesteps within the experience replay, which drives the model to learn features that are consistently predictive through time.

## 8.4 Predictability and Contrastive Learning

Information maximization has long been considered one of the standard principles for measuring correlation and performing feature selection [Song et al., 2012]. In the MDP context, high values of $\mathcal{I}([S_t, A_t], S_{t+k})$ indicate that $(S_t, A_t)$ and $S_{t+k}$ have some form of dependence, while low values suggest independence. The fact that predictability (or more precisely determinism) in Markov systems is linked to the MI suggests a deeper connection to the spectrum of the transition kernel $T$. For instance, the set of eigenvalues of $T$ for a Markov decision process contains important information about the connectivity of said process, such as mixing time or number of densely connected clusters [Von Luxburg, 2007, Levin and Peres, 2017].

Consider the setting in which $\pi$ is fixed at some iteration in the optimization process. In the rest of this section, we let $\mathbf{T}$ denote the expected transition model $\mathbf{T}(s, s') = \mathbb{E}_\pi[T(s, a, s')]$ (it is a Markov chain). We let $\nu_t(s, s') = \frac{\mathbf{T}(s,s')}{p_{t+1}(s')}$ be the ratio learnt when optimizing the infoNCE loss on samples drawn from the random variables $S_t$ and $S_{t+1}$ (for a fixed $t$) [Oord et al., 2018]. We also let $\nu_\infty(s, s') = \frac{\mathbf{T}(s,s')}{\boldsymbol{\rho}(s')}$ be that ratio when the Markov chain has reached its stationary distribution $\boldsymbol{\rho}$ (see Section 8.1), and $\tilde{\nu}_t(s, s')$ be the scoring function learnt using InfoNCE (which converges to $\nu_t(s, s')$ in the limit of infinite samples drawn from $(S_t, S_{t+1})$).

**Proposition 1** *Let $0 < \epsilon \leq 1$. Assume at time step $t$, training of $\tilde{\nu}_t$ has close to converged on a pair $(s, s')$, i.e. $|\nu_t(s, s') - \tilde{\nu}_t(s, s')| < \epsilon$. Then the following holds:*

$$t \geq t_{mix}\left(\frac{\epsilon}{2} \min_x \boldsymbol{\rho}(x)^2\right) \quad \Longrightarrow \quad \left|\nu_t(s, s') - \nu_\infty(s, s')\right| \leq 2\epsilon. \tag{11}$$

**Proof 1** *Let us consider fixed $0 < \epsilon \leq 1$, $(s, s')$ and $t \geq t_{mix}(\frac{\epsilon}{2} \min_x \boldsymbol{\rho}(x)^2)$. First, since $t_{mix}(\frac{\epsilon}{2} \min_x \boldsymbol{\rho}(x)^2) \geq t_{mix}(\frac{\min_x \boldsymbol{\rho}(x)}{2})$, we have*

$$|p_{t+1}^*(s') - \boldsymbol{\rho}(s')| \leq \frac{\min_x \boldsymbol{\rho}(x)}{2}.$$

*Or in other terms: $p_{t+1}^*(s') \geq \frac{\min_x \boldsymbol{\rho}(x)}{2}$. Now, we have:*

$$|\tilde{\nu}_t(s, s') - \nu_\infty(s, s')| \leq |\tilde{\nu}_t(s, s') - \nu_t(s, s')| + |\nu_t(s, s') - \nu_\infty(s, s')|$$

$$\leq \epsilon + \left|\frac{\mathbf{T}(s, s')}{p_{t+1}(s')} - \frac{\mathbf{T}(s, s')}{\boldsymbol{\rho}(s')}\right|$$

$$\leq \epsilon + \frac{|p_{t+1}(s') - \boldsymbol{\rho}(s')|}{p_{t+1}(s')\boldsymbol{\rho}(s')}$$

$$\leq \epsilon + \frac{|p_{t+1}(s') - \boldsymbol{\rho}(s')|}{\frac{\min_x \boldsymbol{\rho}(x)}{2} \min_x \boldsymbol{\rho}(x)}.$$

*By assumption on $t$, we know that $|p_{t+1}(s') - \boldsymbol{\rho}(s')| \leq \frac{\epsilon}{2} \min_x \boldsymbol{\rho}(x)^2$, which concludes the proof.*

Proposition 1 in conjunction with the result on Markov chain mixing times from Levin and Peres [2017] suggests that faster convergence of $\tilde{\nu}_t$ to $\nu_\infty$ happens when the *spectral gap* $1 - \lambda_{(2)}$ of $T$ is large, or equivalently when $\lambda_{(2)}$ is small. It follows that, on one hand, mutual information is a natural measure of concordance of $(s, s')$ pairs and can be maximized using data-efficient, batched gradient methods. On the other hand, the rate at which the InfoNCE loss converges to its stationary value

(ie maximizes the lower bound on MI) depends on the spectral gap of $T$, which is closely linked to predictability. This relation holds in very simple domains like the Markov Chains which were presented across the paper, but for now, there is no reliable way to estimate the second eigenvalue of the MDP transition operator under nonlinear function approximation that we are aware of.

## 8.5 Code snippet for DIM objective scores

The following snippet yields pointwise (i.e. not contracted) scores given a batch of data.

```python
def temporal_DIM_scores(reference,positive,clip_val=20):
    """
    reference: n_batch × n_rkhs × n_locs
    positive: n_batch x n_rkhs x n_locs
    """
    reference = reference.permute(2,0,1)
    positive = positive.permute(2,1,0)
    # reference: n_loc × n_batch × n_rkhs
    # positive: n_locs × n_rkhs × n_batch
    pairs = torch.matmul(reference, positive)
    # pairs: n_locs × n_batch × n_batch
    pairs = pairs / reference.shape[2]**0.5
    pairs = clip_val * torch.tanh((1. / clip_val) * pairs)
    shape = pairs.shape
    scores = F.log_softmax(pairs, 2)
    # scores: n_locs × n_batch × n_batch
    mask = torch.eye(shape[2]).unsqueeze(0).repeat(shape[0],1,1)
    # mask: n_locs × n_batch × n_batch
    scores = scores * mask
    # scores: n_locs × n_batch × n_batch
    return scores
```

To obtain a scalar out of this batch, sum over the third dimension and then average over the first two.

## 8.6 Experiment details

All experiments involving RGB inputs (Ising, Ms.PacMan and Procgen) were ran with the settings shown in Table 2. Parameters such as gradient clipping and n-step-returns were kept from the codebase, 'rlpyt', since it was observed that they helped achieve a more stable convergence.

The global DIM heads consist of a standard single hidden fully-connected layer network of 512 with ReLU activations and a skip-connection from input to output layers. The action is transformed into one-hot and then encoded using a 64 unit layer, after which it is concatenated with the state and passed to the global DIM head.

The local DIM heads consist of a single hidden layer network made of $1 \times 1$ convolution. The action is tiled to match the shape of the convolutions, encoded using a $1 \times 1$ convolutions and concatenated along the feature dimension with the state, after which is is passed to the local DIM head.

In the case of the Ising model, there is no decision component and hence no concatenation of state and action representations is required.

### 8.6.1 AMI of a biased random walk

We see from the formulation of the mutual information objective $\mathcal{I}(S_{t+1}, S_t)$ that it inherently depends on the ratio of $\mathbf{P}/\rho$. Recall that, for a 1-d random walk on integers $[0, N)$, the stationary distribution is a function of $\frac{\alpha}{1-\alpha}$ and can be found using the recursion $\mathbb{P}[S = i] = \alpha\mathbb{P}[S = i - 1] + (1 - \alpha)\mathbb{P}[S = i + 1]$. It has the form

$$\rho(i) = \mathbb{P}[S = i] = r^i(1 - r)(1 - r^N)^{-1}, \ \ i \in [0, N), \tag{12}$$

| Name | Description | Value |
|---|---|---|
| $\varepsilon T_{exploration}$ | Exploration at $t=0$ | 0.1 |
| $\varepsilon T_{exploration}$ | Exploration at $t=T_{exploration}$ | 0.01 |
| $T_{exploration}$ | Exploration decay | $10^5$ |
| LR | Learning rate | $2.5 \times 10^{-4}$ |
| $\gamma$ | Discount factor | 0.99 |
| Clip grad | Gradient clip norm | 10 |
| N-step-return | N-step return | 7 |
| Frame stack | Number of stacked frames | 1 (Ising and Procgen) <br> 4 (Ms.PacMan) |
| Grayscale | Grayscale or RGB | RGB |
| Input size | State input size | $84 \times 84$ (Ising and Ms.PacMan) <br> $80 \times 104$ (Procgen) |
| $T_{warmup}$ | Warmup steps | 1000 |
| Replay size | Size of replay buffer | $10^6$ |
| $\tau$ | Target soft update coeff | 0.95 |
| Clip reward | Reward clipping | False |
| $\lambda_{4t4}$ | Global-global DIM | 1 (Ms.PacMan and Procgen) <br> 0 (Ising) |
| $\lambda_{3t3}$ | Local-local DIM | 1 (Ms.PacMan and Ising) <br> 0 (Procgen) |
| $\lambda_{3t4}$ | Local-global DIM | 0 |
| $\lambda_{4t3}$ | Global-local DIM | 0 |
| $k$ | DIM lookahead constant | 1 (Ising and Ms.PacMan) <br> Variable between 1 and 5 (Procgen) |

Table 2: Experiments' parameters

for $r = \frac{\alpha}{1-\alpha}$.

The pointwise mutual information between states $S_t$ and $S_{t+1}$ is therefore the random variable

$$\begin{aligned}
\dot{\mathcal{I}}(S_{t+1}, S_t) &= \log \mathbb{P}[S_{t+1}|S_t] - \log \mathbb{P}[S_{t+1}] \\
&= \log \alpha^{\mathbb{1}_{(>0)}(S_{t+1}-S_t)} + \log(1-\alpha)^{\mathbb{1}_{(<0)}(S_{t+1}-S_t)} - \log \rho(S_{t+1})
\end{aligned} \tag{13}$$

with expectation equal to the average mutual information which we can find by maximizing, among others, the InfoNCE bound. We can then compute the AMI as a function of $\alpha$

$$\mathcal{I}(S_{t+1}, S_t; \alpha) = \sum_{i=0}^{N} \sum_{j=0}^{N} \dot{\mathcal{I}}(j,i)\alpha^{\mathbb{1}_{(>0)}(j-i)}(1-\alpha)^{\mathbb{1}_{(<0)}(j-i)}\rho(i), \tag{14}$$

which is shown in Figure 3c.

The figures were obtained by training the global DIM objective $\Phi_4$ on samples from the chain for 1,000 epochs with learning rate $10^{-3}$.

### 8.6.2 Ising model

We start by generating an $84 \times 84$ rectangular lattice which is filled with Rademacher random variables $v_{1,1}, .., v_{84,84}$; that is, taking $-1$ or $1$ with some probability $p$. For any $p \in (0,1)$, the joint distribution $p(v_{1,1}, .., v_{84,84})$ factors into the product of marginals $p(v_{1,1})..p(v_{84,84})$.

At every timestep, we uniformly sample a random index tuple $(i,j), 21 \le i,j \le 63$ and evolve the set of nodes $\boldsymbol{v} = \{v_{k,l} : i - 21 \le k \le i + 21, j - 21 \le l \le j + 21\}$ according to an Ising model with temperature $\beta^{-1} = 0.4$, while the remaining nodes continue to independently take the values $\{-1, 1\}$ with equal probability. If one examines any subset of nodes outside of $\boldsymbol{v}$, then the information conserved across timesteps would be close to 0, due to observations being independent in time.

However, examining a subset of $\boldsymbol{v}$ at timestep $t$ allows models based on mutual information maximization to predict the configuration of the system at $t + 1$, since this region has high mutual information across time due to the ratio $\frac{\mathbf{T}(v,v')}{p_{t+1}(v')}$ being directly proportional to the temperature parameter $\beta^{-1}$.

To obtain the figure, we trained local DIM $\Phi_3$ on sample snapshots of the Ising model as $84 \times 84$ grayscale images for 10 epochs. The local DIM scores were obtained by feeding a snapshot of the Ising model at $t = 3$; showing it snapshots from later timestep would've made the task much easier since there would be a clear difference in granularities of the random pattern and Ising models.

### 8.6.3 Ms.PacMan

Figure 6: The simplified Ms.PacMan environment

In PacMan, the agent, represented by a yellow square, must collect food pellets while avoiding four harmful ghosts. When the agent collects one of the boosts, it becomes invincible for 10 steps, allowing it to destroy the enemies without dying. In their turn, ghosts alternate between three behaviours: 1) when the agent is not within line-of-sight, wander randomly, 2) when the agent is visible and does not have a boost, follow them and 3) when the agent is visible and has a boost, avoid them. The switch between these three modes happens stochastically and quasi-independently for all four ghosts. Since the food and boost pellets are fixed at the beginning of each episode, randomness in the MDP comes from the ghosts as well as the agent's actions.

The setup for our first experiment in the domain is as follows: with a fixed probability $\varepsilon$, each of the 4 enemies take a random action instead of following one of the three movement patterns.

The setup for our second experiment in the domain consists of four levels: in each level, only one out of the four ghosts is lethal - the remaining three behave the same but do not cause damage. The model trains for 5,000 episodes on level 1, then switches to level 3, then level 3 and so forth. This specific environment tests for the ability of DIM to quickly figure out which of the four enemies is the lethal one and ignore the remaining three based on color .

For our study, the state space consisted of $21 \times 19 \times 3$ RGB images. The inputs to the model were states re-scaled to $42 \times 38 \times 12$ by stacking 4 consecutive frames, which were then concatenated with actions using an embedding layer.

The third experiment consisted in overlaying the Ising model from the above section onto walls in the Ms.PacMan game. Every rollout, the Ising model was reset to some (random) initial configuration and allowed to evolve until termination of the episode. The color of the Ising distractor features was chosen to be fuchsia.

### 8.6.4 Procgen

The training setting consists in fixing the first 500 levels of a given Procgen game, and train all algorithms on these 500 levels in that specific order. Since we use the Nature architecture of DQN rather than IMPALA (due to computational restrictions), our results can be different from other Procgen baselines.

The data augmentation was tried only for DRIML-fix - DRIML-ada seems to perform well without data augmentation. The data augmentation steps performed on $S_t$ and $S_{t+k}$ fed to the DIM loss

Figure 7: Upsampled screen cap of the Ms.PacMan task with Ising distractor features (in fuchsia).

consisted of a random crop ($0.8$ of the original's size) with color jitter with parameters $0.4$. Although the data augmentation is helpful on some tasks (typically fast-paced, requiring a lot of camera movements), it has shown detrimental effects on others. Below is a list of games on which data augmentation was beneficial: bigfish, bossfight, chaser, coinrun, jumper, leaper and ninja.

The $k$ parameter, which specifies how far into the future the model should make its predictions, worked best when set to 5 on the games: bigfish, chaser, climber, fruitbot, jumper, miner, maze and plunder. For the remaining games, setting $k = 1$ yielded better performance.

**Baselines** The baselines were implemented on top of our existing architecture and, for models which use contrastive objectives, used the exactly same networks for measuring similarity (i.e. one residual block for CURL and CPC). CURL was implemented based on the authors' code included in their paper and that of MoCo, with EMA on the target network as well as data augmentation (random crops and color jittering) on $S_t$ for randomly sampled $t > 0$.

The No Action baseline was tuned on the same budget as DRIML, over $k = 1, 5$ and with/without data augmentation. Best results are reported in the main paper.

| | DRIML-noact (k=1) | DRIML-noact (k=5) |
|---|---|---|
| bigfish | $1.193 \pm 0.04$ | $1.33 \pm 0.12$ |
| bossfight | $0.466 \pm 0.07$ | $0.472 \pm 0.01$ |
| caveflyer | $8.263 \pm 0.26$ | $5.925 \pm 0.18$ |
| chaser | $0.224 \pm 0.01$ | $0.229 \pm 0.02$ |
| climber | $1.359 \pm 0.13$ | $1.574 \pm 0.01$ |
| coinrun | $13.146 \pm 1.21$ | $9.632 \pm 2.8$ |
| dodgeball | $1.221 \pm 0.04$ | $1.213 \pm 0.09$ |
| fruitbot | $0.714 \pm 0.31$ | $5.425 \pm 1.33$ |
| heist | $1.042 \pm 0.02$ | $0.861 \pm 0.07$ |
| jumper | $2.966 \pm 0.1$ | $4.314 \pm 0.64$ |
| leaper | $5.403 \pm 0.09$ | $3.521 \pm 0.3$ |
| maze | $0.984 \pm 0.13$ | $1.438 \pm 0.26$ |
| miner | $0.11 \pm 0.01$ | $0.116 \pm 0.01$ |
| ninja | $6.437 \pm 0.22$ | $5.9 \pm 0.38$ |
| plunder | $2.67 \pm 0.08$ | $3.2 \pm 0.05$ |
| starpilot | $3.699 \pm 0.3$ | $2.951 \pm 0.31$ |

Table 3: Ablation of the impact of predictive timestep in NCE objective (i.e. $k$) on the no action model's training performance (50M training frames).

## Footnotes

[8]The existence and uniqueness of $\boldsymbol{\rho}$ are direct results of the Perron-Frobenius theorem.

[9]We consider discrete state and action spaces.