[Reviews · NeurIPS 2020]

Review 1

Summary and Contributions: The paper studies the effect of using an unsupervised loss which recently gained popularity in representation learning (InfoNCE) as an auxillary loss in RL, in order to bias the representations learned by the model to maximise mutual information between local views (features in the early layers of the network) and global view (features in later layers), loosely inspired by DIM (Hjelm et al.). The authors contributions consist of - formulation of a InfoNCE loss for triplets of current state, action,successor states - - evaluation of this loss in this supervised setting on learning the transition model of a biased random walk chains and learning to predict the deterministic sections of patches of Ising models embedded in random pixel noise, as well as in 2 continous RL learning setting based on Ms Pacman and an evaluation on procgen, comparing to C51, CURL (another work using a similar auxiliary loss) - - a theoretic contribution about the link between the stationary distribution linking the convergence of the InfoNCE loss to the environment distribution, which is tested in an experiment

Strengths: + it applies a promising direction from representation learning to RL and achieves decent results + it presents an interesting theoretical insight as support for using the InfoNCE loss, which seems to hold predictive power confirmed in an experiment + it is for the most part clearly structured and well written and cites the relevant literature as known to me + Good representations in Deep RL seem to be a key for improving performance and sample complexity, so the relevance is a give

Weaknesses: - It achieves some improvements over the C51 baseline, but not groundbreaking ones, which in combination with the prior work could turn this into an incremental improvement. The theoretical insight helps against this, but is only a small part of the paper. - It draws some parallels to the DIM paper without (as it seems to me from reading the paper and the supplementary) using a key component of the DIM which is the local-MI maximisation *in combination* with the global MI and a prior term. If I misunderstood the authors this should be made clearer in equation 4,5 and 7 - - In the same vein, the MI maximisation is performed between layers 3 and 4, not between the input and the higher layers as in DIM. While I can think of justifications for this given we are doing RL here and not unsupervised representation learning, I'd prefer the authors do so and make them clear in the paper (see also my bullet points below) - I am missing a comparison with the closest work I know to this (DeepMDP and CURL), especially in terms of what biases their loss terms express - the broader impact is less about this work than about the field of deep RL in general Edit:The most important (comparison with CURL,DeepMDP; deterministic noise experiment) have been addressed

Correctness: I have gone over the claims and evaluation and as best I can tell, the derivations are correct and the methodology employed seemed sound. I have not examined the code.

Clarity: Yes, except for legibility issues mentioned it is clear and readable

Relation to Prior Work: The relevant previous work is mentioned, but I am missing a deeper comparison with CURL and DeepMDP

Reproducibility: Yes

Additional Feedback: As it is, I think this paper is *barely* a weak reject. I would give the following recommendations to upgrade it to an accept: - perform an ablation study comparing the effect of choosing earlier layers as local features (i.e. the input itself, as well as the outputs of f_1 and f_2) and discuss the final choice of local features/results - if comparing with the DIM, there should be a discussion of the local MI, as it is there is only the simple InfoNCE objective maximising global mutual information between layers - add an ablation study that features "deterministic noise" processes (e.g. a patch of ising model development in the walls of the pacman observations) to see if the performance is strongly negatively impacted - the distinction between this work and DeepMDP, CURL etc. should be made clearer - reword the broader impact with more language sepcifically about this paper, not about deep RL in general More experiments and a more in depth exploration of the theory would lead to a strong accept as the DNA of the work is good, but I assume the above is all that is feasible for the current venue. # Other Minor comments - line 148 "residual and CNN" I assume this is meant to say "a residual CNN" - plots/charts barely legible, would add larger copy in supplementary - adding some annotation/comment for the proof will help the casual reader, even if not strictly necessary here # Edit The most important (comparison with CURL,DeepMDP; deterministic noise experiment) have been addressed, but not all (comments on feature selection MI). Other reviewers have noted that the paper could benefit from another go before being submitted at another venue. I will therefore increase my score to a marginal accept instead of a clear accept.


Review 2

Summary and Contributions: The paper introduces a new auxiliary task for deep reinforcement learning that maximizes the mutual information between two temporally-related observations. This is done by using noise contrastive estimation (NCE) to tell apart positive and negative samples of future observations. This auxiliary task is motivated by the hypothesis that representations that are predictive of future observations are better suited to adapting to new environments than representations that are predictive of the reward signal alone. Finally, the paper provides empirical evaluations that demonstrate that (1) the auxiliary task is able to approximate the mutual information between consecutive states in a Markov chain, (2) the main algorithm has improved performance in a continual learning version of Ms. Pacman, and (3) the auxiliary task may improve the performance of C51 in complex environments. The three main contributions of the paper are: 1. Algorithmic: Proposes a new auxiliary task for deep RL with a corresponding loss function and neural network architecture. A novel aspect of this architecture is the incorporation of the action in the NCE objective. 2. Theoretical: The paper alludes to a connection between the mixing time of the Markov chain induced by a policy and the convergence rate of the NCE loss used for the auxiliary task. 3. Empirical: Provides empirical evaluation of the main algorithm, Deep Reinforcement and Information Learning or DRIML, in a continual learning task and several complex environments. Moreover, the paper provides evidence that the auxiliary task is useful for estimating the mutual information between consecutive states in a Markov Chain.

Strengths: The deep information maximization objective combined with noise contrastive estimation (InfoNCE) is a fairly new unsupervised learning loss that has yet to be thoroughly explored in deep reinforcement learning. The main value of the paper is the study of the representations learned when optimizing the InfoNCE loss and how those representations can be used for continual learning. Moreover, the paper introduces a novel architecture that uses the action information as part of the InfoNCE loss. These two ideas are novel and, to my knowledge, they haven’t been presented in the literature before. In terms of significance, there has been growing interest in the representations learned by the InfoNCE loss in the context of reinforcement learning; see, Oord, Li, and Vinyals (2018), Anand et. al. (2019), and Laskin, Srinivas, and Abbeel (2020). Thus, given the novelty of the ideas in the paper, I believe the paper would be of interest to this community. References: van den Oord, A., Li, Y., & Vinyals, O. (2018). Representation learning with contrastive predictive coding. CoRR,abs/1807.03748arXiv 1807.03748. http://arxiv.org/abs/1807.03748 Anand, A., Racah, E., Ozair, S., Bengio, Y., Cˆot ́e, M.-A., & Hjelm, R. D. (2019). Unsuper-vised state representation learning in atari (H. Wallach, H. Larochelle, A. Beygelzimer,F. d’Alch ́e-Buc, E. Fox, & R. Garnett, Eds.). In H. Wallach, H. Larochelle, A. Beygelz-imer, F. d’Alch ́e-Buc, E. Fox, & R. Garnett (Eds.), Advances in neural information processing systems 32. Curran Associates, Inc. http://papers.nips.cc/paper/9081-unsupervised-state-representation-learning-in-atari.pdf. Laskin, M., Srinivas, A., & Abbeel, P. (2020). Curl: Contrastive unsupervised representations for reinforcement learning [arXiv:2003.06417]. Proceedings of the 37th International Conference on Machine Learning, Vienna, Austria, PMLR 119. References for the next section (I ran out of space): Hessel, M., Modayil, J., van Hasselt, H., Schaul, T., Ostrovski, G., Dabney, W., Horgan, D.,Piot, B., Azar, M., & Silver, D. (2018). Rainbow: Combining improvements in deep reinforcement learning, In AAAI conference on artificial intelligence. https://www.aaai.org/ocs/index.php/AAAI/AAAI18/paper/view/17204/16680. Lillicrap, T., Hunt, J., Pritzel, A., Heess, N., Erez, T., Tassa, Y., Silver, D., & Wierstra, D.(2016). Continuous control with deep reinforcement learning, In International conference of learning representations. https://arxiv.org/pdf/1506.02438.pdf.

Weaknesses: The main weakness of the paper is its lack of focus, which is most evident in empirical evaluations and theoretical results that don’t seem relevant to the main ideas of the paper. I don’t think this is because the empirical and theoretical results are not relevant, but because the paper emphasizes the wrong aspects of these results. To reiterate, the main idea of the paper is that the representations learned when minimizing the InfoNCE loss may be useful for continual learning in cases where the environment dynamics don’t change too much but the reward function does. A secondary idea is the addition of the action information to the InfoNCE. About the last experiment in the procgen environment (Section 6.4), the section reads as an attempt to demonstrate the main algorithm (DRIML) is the best. Not only is this not true because nothing conclusive can be said with such few runs, but it obfuscates more interesting findings and relevant information. - First, it would be useful to provide some relevant information about why these evaluations were performed in the procgen environment. This choice is important for the main hypothesis because procgen environments are procedurally generated. Hence, if we hypothesize that DRIML will learn a robust representation that captures the environment dynamics and will be better suited to overcome the random variations in the environment, then we would expect DRIML to perform better than other models that are not explicitly designed this way, such as C51. This is indeed what happens, but the text does not emphasize what the main hypothesis is and why this environment is relevant. - Second, there are some interesting findings that are not emphasized enough in Table 1. The impact of the action information on the performance of DRIML is striking. In some environments such as jumper, the performance almost tripled. Additionally, it is possible that the advantage that DRIML has over CURL is due to the action information. Here, it would be good to emphasize this fact and leave it for future work to investigate whether CURL would benefit from including the action information into its architecture. These two additions would make the argument stronger because instead of a simple comparison to determine which algorithm is best, the emphasis would be on the two main ideas of the paper that motivate the DRILM agent. About the first and second experiments (Section 6.1 and 6.2), these three sections are great for demonstrating that DRIML is indeed working as intended. However, it is often difficult to tell what is the main takeaway from each experiment because the writing doesn’t emphasize the appropriate parts of the experiments. - In Section 6.1, it seems that the wrong plots are referenced in Lines 217 and 218. The paragraph references FIgure 2b and 2c, but it should be referencing 2a and 2b. Moreover, it would be useful to have more details about these two plots: what are the x and y axis, what would we expect to see if DRIML was working as intended, and why do the plots have different scales? For Figure 2c, it is not clear why it is included. It seems to be there to justify the choice of alpha = 0.499; if this is the case, it should be explicitly stated. Figure 2d is never referenced and it’s not clear what the purpose of this figure is, so it should be omitted. - In Section 6.2, it isn’t clear what architecture is used in the experiment and how the DIM similarity is computed. An easy fix for this is to move most of the information about the Ising model from the main text to the appendix (Section 8.6.1) and move the information about the architecture to the main text. In fact, the appendix motivates this experiment fairly well in Lines 511 to 513: “If one examines any subset of nodes outside of [a patch], then the information conserved across timesteps would be close to 0, due to observations being independent in time.” You can motivate the hypothesis of this experiment based on this statement: if the DIM loss in Equation (6) is measuring mutual information across timesteps, then we would expect its output to have high measure inside of the patches and a low measure outside of the patches. This would make it very clear that the DIM loss is in fact working as intended. About the theoretical results, the main issue is the organization and the lack of connection between the theoretical results and the main ideas of the paper. - In terms of organization, it seems odd that Theorem 3.1 is introduced in page 3, but is referenced until page 6 after Proposition 1. It would be easier on the reader to have these two results close together. - More importantly, it is not clear what the connection between the theoretical results and the main idea of the paper is. The proposition is used as evidence that the convergence rate of \tilde{ v_t } is proportional to the second eigenvalue of the Markov Chain induced by the policy. However, I don’t follow the logic used for this argument since the proposition tells us that if \tilde{ v_t } and v_t are close, then v_t and v_\infty are also close. Combined with Theorem 3.1, this tells us that v_t will converge to v_\infty in a time proportional to the second eigenvalue of the Markov Chain and the error between v_t and \tilde{ v_t }, but it says nothing about the convergence rate of \tilde{ v_t } to v_t. Even if this was true, it is not clear how this convergence rate relates to the continual learning setting, which is the motivating problem of the paper. One could make a connection by arguing that in environments where the dynamics don’t change but the reward function does, the convergence rate of the InfoNCE loss remains unchanged. However, this is not what is written in the paper. This, in my opinion, is the weakest part of the paper, to the point where the paper would be better off without it since it is not adding much to the main argument. Perhaps this proof would be better suited for a different paper that specifically focuses on the convergence rate of the InfoNCE loss. Finally, there are a few architectural choices that are not well motivated. -It is not clear why the algorithm uses 4 different auxiliary losses: local-local, local-global, global-local, and global-global in Equation (7). To justify this choice, it would be useful to have an ablation study that compares the performance of DRIML with and without each of these losses. - Second, in Algorithm Box 1, it is not clear why each auxiliary loss is optimized separately instead of optimizing all of them at once. - Third, it’s not clear what architecture is used for the DRIML agent. Line 11 in the abstract mentions that the paper augments the C51 agent, but line 259 says that “all algorithms are trained… with the DQN... architecture,” yet Table 2 in the appendix (Section 8.5) shows hyperparameters that are not part of the DQN or C51 architectures. For example, gradient clipping, n-step returns, and soft target updates (tau in Table 2) are not original hyperparameters of the DQN or C51 architectures. The n-step return is more commonly associated with the Rainbow architecture (Hessel et. al., 2018) and the soft target updates correspond to the continuous control agent from Lillicrap et. al. (2016). There should be some explanation about these choices and, more importantly, the paper should clarify whether the other baselines also use these modifications. Of particular interest to me is the motivation behind gradient clipping since it is not used in any of the 4 architectures mentioned above; is this essential for the DRIML agent? - Finally, how were all these hyperparameters selected? Neither the main text or the appendix provide an explanation for these choices of hyperparameter values.

Correctness: I checked the validity of the proof of Proposition 1. The algebra checks out, but there’s one important assumption that is not emphasized enough: the convergence of \tilde{ v_t } to v_t. The proposition relies on the fact that \tilde{ v_t }, the function learned when minimizing the InfoNCE loss, is closed to convergence to v_t; however, this is not a trivial assumption since there’s no existing proof that \tilde{ v_t } converges to v_t when the data generating process is a Markov chain. This is another reason why I consider this is not the right paper for Proposition 1; the argument that the paper is trying to make requires a more nuanced and complete analysis than the analysis currently presented.

Clarity: Overall, the writing is clear. However, there are some claims that are not completely accurate or clear, which I list below: - Line 22-24: “Model-free methods … focus on learning a policy that maximizes reward or a function that estimates the optimal value of states and actions.” This is true, but model based methods may also learn policies and value functions from states and rewards sampled from the model, see Dynamic Programing in Sutton and Barto (2018). It would be more accurate to say that policies and value functions are learned directly from observations sampled by the agent as it is interacting with the environment. - Line 55: “Humans are able to retain old skills when taught new ones [Wixted, 2004].” This claim does not require a citation. Nevertheless, if the paper is to reference Wixted’s (2004) work, It would be better to include a more specific statement that is more closely related to Wixted’s work. - Line 147: The definitions of “local” and “global” features are not clear unless the reader is fully familiar with the work of Hjelm et. al. (2019). The paper should provide some context about these two terms. - Line 152-153: “train the encoder to maximize mutual information (MI) between local and global ‘view’.” This is not true. The encoder is trained to minimize the loss function in Equation (4), not to maximize MI. - Line 192-193: “\tilde{ v }_t (s, s’) be the scoring function learnt using InfoNCE (which converges to v_t (s, s’) in the limit of infinite samples drawn from (S_t, S_{t+1}).” To my knowledge, there’s no formal proof of this; if there is, then it should be cited here. - Line 217-218: “Results are shown in Fig. 2b, they are well aligned with the true transition matrix (Fig. 2c).” It should be Fig. 2a instead of Fig. 2c. Moreover, what does “well aligned” mean in this context? It would greatly improve clarity if the statement was more specific. - In Figure 3 it is not clear what each of these plots corresponds to. The caption mentions that the grayscale plots correspond to 3 systems at t = 32, while the color plots correspond to the DIM similarity score between t =2 and t = 3. However, in Lines 231-232, the text mentions that Figure 3 corresponds to the system at t =32 during three different episodes. Yet, in Lines 234-235 the text mentions that Figure 3 corresponds to the local features of states t = 2 and the same features at t = 3. Which one is it? This is very confusing and I could not fully determine what was going on. - Line 271: “DRIML implicitly learns a transition model by boosting state similarity.” This is not what is shown in the paper. The DRIML agent implicitly learns to maximize the mutual information between consecutive representations through minimizing the InfoNCE loss; however, this does not imply that the DRIML agent is implicitly learning a transition model of the environment. - Line 272-273: “[DRIML] boosts training performance in complex domains such as Procgen.” There is not sufficient evidence for this since this claim is based on very few runs. It would be better to limit the scope of the conclusions to the main ideas of the paper. - In the appendix, Line 475 and 478, what is the definition of v* and p*? References: Sutton, R. S., & Barto, A. G. (2018). Reinforcement learning: An introduction. Cambridge,MA, USA, A Bradford Book. Hjelm, R. D., Fedorov, A., Lavoie-Marchildon, S., Grewal, K., Bachman, P., Trischler, A., & Bengio, Y. (2018). Learning deep representations by mutual information estimation and maximization. arXiv preprint arXiv:1808.06670.

Relation to Prior Work: The literature review is thorough. The only improvement that I could suggest is to emphasize the differences between the DRIML agent and the CURL agent, i.e., the incorporation of the action information and the use of loca-local and global-global losses.

Reproducibility: Yes

Additional Feedback: Although the paper presents some valuable work, I don't think the paper is ready for publication in its current form. Nevertheless, I believe most of my comments can be addressed before the camera-ready deadline, and I’d be willing to increase my score based on the author’s rebuttal and the discussion with other reviewers. === Final Comments === I appreciate that the authors were receptive about most of our comments and hope that this will help make a stronger submission. However, since it seems that substantial changes will be done to the paper, I think it would be better for the paper to have another set of reviewers read through it and provide feedback. Thus, I have kept the same score. I believe the paper is very close to publication standards and that it has valuable contributions, so I encourage the authors to keep working on it.


Review 3

Summary and Contributions: This paper uses contrastive representation learning and data augmentation as an auxiliary objective to provide improved representation learning in reinforcement learning agents. The form of the contrastive loss encourages maximizing the mutual information between ‘local’ and ‘global’ representations (i.e. low-level and high-level representations within the network), for both the current state-action and future states. This auxiliary objective is demonstrated to produce reasonable representations of similarity on a small problem (Ising model). Next, the authors show (in a simplified MsPacMan-style game) that, when equipped with the Deep InfoMax (DIM) objective, an agent is better able to adapt to changes in the environment (changing which ghost is deadly). Finally, they combine the DIM objective with the C51 agent and evaluate performance on a suite of procedurally generated domains, finding improved performance over C51 and a recent contrastive method (CURL).

Strengths: The empirical work in this paper is one of its primary strengths. The early experiments demonstrating properties of the learned representations (transition ratio, Ising model experiments) show that (in small environments) some of the aims motivating the DIM objective are indeed achieved. The next experiment, to me, was particularly interesting. It showed (in the simplified MsPacMan game) that, perhaps due to capturing transition dynamics, the representation allowed faster transfer to a changing environment. Finally, the results on the Procgen benchmark show that in larger problems the method continues to confer an advantage over both an existing RL agent and a recent contrastive learning method.

Weaknesses: The writing and structure of the paper could be improved. While the introduction and background had a coherent narrative, the remaining sections feel largely disjoint and do not lead naturally from one to the next. There are areas which are essential to the work but received very limited detail (I’m thinking in particular of Section 4). There have been a number of recent works studying the use of contrastive representation learning and data augmentation. I think the authors did not do an entirely satisfactory job of differentiating their contributions from that of existing work. Expanding section 4 to clarify their use of the DIM objective, and later being more explicit about what form of it is used for what experiments, would have help distinguish this work from others.

Correctness: Regarding correctness, I only had one concern: the experiments with procgen environments require more detail about the experimental setup. How are the sequence of levels generated? Is the reported performance on the training or testing environments? I’m assuming that this was done with random sequences of levels and reporting test performance, but this needs to be said explicitly.

Clarity: As mentioned above, while the writing itself is not ‘bad’ it could be significantly improved to be clearer, with a focus on putting results into context and connecting the different sections together. A great example of these problems comes in Section 3.1. The authors present Theorem 3.1, but it is not discussed or introduced in any way and its presence is only understood a few pages later when Proposition 1 is given.

Relation to Prior Work: The background section does a good job of discussing related work, but does not do a very good job of *contrasting* the proposed approach with that of similar existing methods.

Reproducibility: No

Additional Feedback: The appendix mention CPC in a way that leads me to wonder if there might have been experimental results which used CPC as a baseline. If so, I think this is something that should be included. Even before getting to this point, I had wanted to see both CURL and CPC as baselines to compare with. I do not think that simply reporting final performance is very informative for RL agents/domains. Because of this, I’d suggest including the learning curves for DRIML and the baselines in the Supplementary Materials. In your rebuttal, could you please precisely explain the key differences between DRIML and CURL/CPC? The most obvious one would be the use of the DIM objective between local/global representations, but since the section explaining this was fairly minimal it would help me see the contributions to be more explicit. Despite my somewhat negative tone, I thought this paper was fairly borderline. ----- Update ------ Thank you for your clear rebuttal, after reading it and discussion with other reviewers I am slightly increasing my score, agreeing with others that what is needed for acceptance is essentially down to editing/writing based on the reviews/rebuttal.


Review 4

Summary and Contributions: This paper learns a representation learning method where the MI between future states and the current state-action tuple are maximized. This would learn and capture a representation that is aware of the transition dynamics and predictive of the future. The proposed method can bring an improvement on a continual learning task where prediction of local dynamics (e.g. moving objects) can be important, and conducts an analysis on the representation in connection to the Markov transition model.

Strengths: Recently there have been many efforts in unsupervised state representation learning techniques for RL, but their main promise is that efficient state representation can benefit downstream RL tasks of learning policies. This paper studies this question, which has certain significance and importance to the community. The introduction tells a clear motivation and story about this. This paper clearly presents interesting analyses on the learned representation in connection to the dynamics of MDP. For example, the fact that DIM implicitly learns a transition matrix (6.1) and discovers a deterministic part demonstrates the value of MI-based representation learning; it also opens up many interesting directions or questions about how the long-term future can be better captured using a similar MI objective. The proposed method has a clear benefit in terms of capturing predictable dynamics than the vanilla model-free RL baseline, showing faster adaptation ability on continual learning setups. This sounds like an interesting result that is closer to the real-world RL settings we should pay more attention to.

Weaknesses: One limitation is that the benchmark of learning RL tasks is only focusing on the Proc-gen benchmark. It would be nice to include more common benchmarks such as Atari or pixel-based continuous control tasks, although they might not be about continual learning settings, to see whether the presented approach is also beneficial in a general setup or more various multi-task setups as well.

Correctness: The proposed representation learning approach is based on maximization of a “future-predictive” state-action mutual information; MI( s_t, a_t ; s_{t+k}). It is a natural but novel idea to consider the joint mutual information of state-action and the future states. The experiments are designed carefully and makes a lot of sense. One remaining question is how sensitive the proposed objective is to the value of k. For example, $k = 1$ (only the next state). An ablative study (i.e. comparing k=1 vs k=5) would be needed.

Clarity: The paper is overall clearly written and easy to follow. A motivation of the problem is good enough, clearly stating the significance of the problem. The subscript notation 3, 4 is actually not very intuitive. I would suggest using ‘L’ (local) or ‘G’ (global), as 3, 4 are choices that are specific to the 5-layer auto-regressive net.

Relation to Prior Work: There is a comprehensive discussion on contrastive-style representation learning in supervised learning and in reinforcement learning context.

Reproducibility: Yes

Additional Feedback: Minor: L167 Figure 4 -> Figure 1

[Author Response · NeurIPS 2020]

We would like to thank all reviewers; the comments are very helpful and we believe that we have addressed the concerns. The modifications have led to a much improved paper. Below we provide a point-by-point response.

**Additional result: Adaptive DRIML** In our original submission, we fixed the temporal offset $k$ of DRIML and found that performance varied across games (see Procgen results in Table 4). This supports a hypothesis that different games express their dynamics at different time scales, so it might be important to look farther ahead (larger $k$) in DRIML for slower games. We learn an "adaptive" $k$ (dependent on the game) directly from the RL policy, using a simple single-hidden-layer neural network trained with SGD that predicts the next action conditioned on the previous action given the current policy. During representation learning, given the sequence of actions from the same trajectory we are drawing states (i.e., the buffer), we sample the length of the sequence of actions $k$ under this transition model by sequential sampling as a Markov process. As can be seen from our results below, DRIML-ada outperforms DRIML-fixed, achieving nearly double the average score of C51. While this modification is small in terms of the auxiliary loss (it learns $k$ without modifying the core algorithm), this addition is novel, insightful, and useful to performance.

**[Differences with and comparison to CURL and CPC]** DRIML encourages consistency in state representations across multiple time steps conditioned on the action through a contrastive loss between pairs of global and local (corresponding to a patches) state representations at different time steps (offset by k time steps). CURL only does contrastive learning between augmented copies of the *same* global state, and does so by relying only on data augmentation and without action-conditioning. We tested data augmentation a la CURL on DRIML (results in Table 3), on some games perf improved, not so much on others. Like DRIML, CPC also predicts future states, but it does not use action-conditioning and relies on autoregression modeled by an RNN. We include additional results below, which are from the best agents on 3 seeds over 50M timesteps: our implementation of 5-step CPC; DRIML-fix and DRIML-ada; CURL (using their code); and DRIML-noact (without actions). All baselines were tuned with the same computation budget. The last row shows the cumulative score over all 16 games normalized by max-min performance as in Cobbe et al. (2019), and wrt C51. We conclude that the additions from DRIML are essential to achieving good performance on ProcGen, as DRIML is the only method that significantly outperforms C51 alone.

**[PacMan + Ising]** As suggested by R1, we also ran DRIML-fix and C51 on our simple PacMan game augmented with an Ising model in irrelevant parts of the screen (e.g. walls). The figure below shows the training returns over 8,000 episodes. This further supports already-established conclusions from the paper.

**[DRIML and DeepMDP]** DeepMDP is an interesting representation-learning method that learns latent reward and transition models from a bisimulation perspective. An important property of maximal bisimulations is that states from which the optimal action sequences are identical are merged. This will make the DeepMDP encoder ignore all state information irrelevant to the optimal policy. In our PacMan experiment, the embedding will not include any knowledge about the harmless ghosts (as they do not impact the actions to take). Upon change of the harmful ghost's color, the encoder will thus need extra samples to adapt. We will expand our discussion of DeepMDP.

**[Other comments]** 1) We report performance on Procgen instead of Atari / MuJoCo as Procgen allows for extremely fast procedural generation of multiple levels, and is better suited for testing continual learning/multi-level setups. 2) We will move their theory to the Appendix as we agree they are not an essential component of the paper. 3) The choice of $\alpha = 0.499$ for the MC experiment is to approach $0.5$ (lowest MI possible) while allowing the matrix to be invertible. Fig. 2d shows that for this class of MC, the spectral gap encodes the "predictivity" of the system, and hence infoNCE is expected to converge faster on $\alpha$ values further from $0.5$. 4) Our implementation relies on `rlpyt` (Stooke and Abbeel, 2019), parameters such as gradient clipping, n-step returns and soft updates were taken to be default values in that library. In Fig.3, the similarity scores are computed between $t = 2$ and $t = 3$, but the grayscale plots show $t = 32$ for visibility (at $t = 2$ the difference between random noise and Ising is barely visible). 5) The reviewers correctly pointed out some typos and presentation issues (e.g. font size), which will be fixed in the next version.

Training performance of DRIML (LL) vs C51 under deterministic Ising noise

| Env | C51 | CPC-1→ 5 | CURL | DRIML-noact | DRIML-fix | DRIML-ada |
|---|---|---|---|---|---|---|
| bigfish | 1.33±0.12 | 1.17±0.16 | 2.7±1.3 | 1.19±0.04 | 2.02±0.18 | **4.45±0.71** |
| bossfight | 0.57±0.05 | 0.52±0.07 | 0.6±0.06 | 0.47±0.01 | 0.67±0.02 | **1.05±0.19** |
| caveflyer | 9.19±0.29 | 6.4±0.56 | 6.94±0.25 | 8.26±0.26 | **10.18±0.41** | 6.77±0.04 |
| chaser | 0.22±0.04 | 0.21±0.02 | 0.35±0.04 | 0.23±0.02 | 0.29±0.02 | **0.38±0.04** |
| climber | 1.68±0.1 | 1.71±0.11 | 1.75±0.09 | 1.57±0.01 | **2.26±0.05** | 2.2±0.08 |
| coinrun | **29.7±5.44** | 11.4±1.55 | 21.17±1.94 | 13.15±1.21 | 27.24±1.92 | 22.88±0.4 |
| dodgeball | 1.2±0.08 | 1.05±0.04 | 1.09±0.04 | 1.22±0.04 | 1.28±0.02 | **1.44±0.06** |
| fruitbot | 3.86±0.96 | 4.56±0.93 | 4.89±0.71 | 5.42±1.33 | 5.4±1.02 | **9.53±0.29** |
| heist | 1.54±0.1 | 0.93±0.08 | 1.06±0.05 | 1.04±0.02 | 1.3±0.05 | **1.89±0.02** |
| jumper | **13.22±0.83** | 2.28±0.44 | 10.27±0.61 | 4.31±0.64 | 12.64±0.64 | 12.16±0.42 |
| leaper | 5.03±0.14 | 4.01±0.71 | 3.94±0.46 | 5.4±0.09 | 6.17±0.29 | **6.35±0.46** |
| maze | 2.36±0.09 | 1.14±0.08 | 0.82±0.2 | 1.44±0.26 | 1.38±0.08 | **2.62±0.1** |
| miner | 0.13±0.01 | 0.13±0.02 | 0.1±0.0 | 0.12±0.01 | 0.14±0.01 | **0.19±0.02** |
| ninja | **9.36±0.01** | 6.23±0.82 | 5.84±1.21 | 6.44±0.22 | 9.21±0.25 | 8.74±0.28 |
| plunder | 2.99±0.07 | 3.0±0.06 | 2.77±0.14 | 3.2±0.05 | 3.37±0.17 | **3.58±0.04** |
| starpilot | 2.44±0.12 | 2.87±0.05 | 2.68±0.09 | 3.7±0.3 | **4.56±0.21** | 2.63±0.16 |
| Norm.score | 1.0 | 0.23 | 0.52 | 0.59 | 1.48 | 1.9 |



[Meta-Review · NeurIPS 2020]

This paper proposes a method to apply noise contrastive estimation for future state prediction as an auxiliary task for RL agents. The authors clearly explain their formulation and through toy experiments show it working as intended. There are some empirical improvements in performance in simple continual learning settings and also in Procgen. The author response contains very useful ablation studies and connection to prior work which I hope the authors consider adding to the final draft, as well as acknowledgement of moving theory sections to make exposition clearer.